# Generation of Functional Vascular Endothelial Cells and Pericytes from Keratinocyte Derived Human Induced Pluripotent Stem Cells

**DOI:** 10.3390/cells10010074

**Published:** 2021-01-05

**Authors:** Selin Pars, Kevin Achberger, Alexander Kleger, Stefan Liebau, Natalia Pashkovskaia

**Affiliations:** 1Institute of Neuroanatomy & Developmental Biology (INDB), Eberhard Karls University Tübingen, Österbergstrasse 3, 72074 Tübingen, Germany; selin.pars@uni-tuebingen.de (S.P.); Kevin.Achberger@uni-tuebingen.de (K.A.); Stefan.liebau@uni-tuebingen.de (S.L.); 2Department of Internal Medicine 1, Ulm University Hospital, 89081 Ulm, Germany; alexander.kleger@uniklinik-ulm.de

**Keywords:** human pluripotent stem cells, hiPSC-derived endothelial cells, hiPSC-derived pericytes, vasculature-on-a-chip, self-assembled microvascular network

## Abstract

Human induced pluripotent stem cell (hiPSC)-derived endothelial cells (ECs) and pericytes provide a powerful tool for cardiovascular disease modelling, personalized drug testing, translational medicine, and tissue engineering. Here, we report a novel differentiation protocol that results in the fast and efficient production of ECs and pericytes from keratinocyte-derived hiPSCs. We found that the implementation of a 3D embryoid body (EB) stage significantly improves the differentiation efficiency. Compared with the monolayer-based technique, our protocol yields a distinct EC population with higher levels of EC marker expression such as CD31 and vascular endothelial cadherin (VE-cadherin). Furthermore, the EB-based protocol allows the generation of functional EC and pericyte populations that can promote blood vessel-like structure formation upon co-culturing. Moreover, we demonstrate that the EB-based ECs and pericytes can be successfully used in a microfluidic chip model, forming a stable 3D microvascular network. Overall, the described protocol can be used to efficiently differentiate both ECs and pericytes with distinct and high marker expression from keratinocyte-derived hiPSCs, providing a potent source material for future cardiovascular disease studies.

## 1. Introduction

Endothelial cells (ECs) represent the main component of blood vessels and play a major role in the regulation of vital physiological processes such as oxygen and nutrient supply to tissues, inflammation, and blood pressure regulation. Additional cell types—pericytes and smooth muscle cells—contribute to vessel dynamics and maturation. Pericytes cover the vascular tube formed by ECs and are essential for the regulation of the blood vessel diameter, tightness of ECs, blood vessel growth, and stability [1]. The pericyte loss is associated with many cardiovascular disorders, as well as edema, diabetic retinopathy, brain hemorrhage, and even embryonic lethality [2].

Fast developing tissue engineering, personalized drug testing, and cardiovascular disease modelling require a robust and easily accessible vascular component: ECs and pericytes, which together form the functional blood vessel. For a long time, human umbilical vein ECs (HUVECs) or primary ECs have served as the most prevalent EC source. However, these cells are not the optimal solution for personalized approaches, because they are not available for the majority of individuals.

Human induced pluripotent stem cells (hiPSCs) can be easily derived from somatic cells and subsequently differentiated to any cell type, providing an alternative, personalized source of vascular cells. Among cell sources for hiPSCs, keratinocytes proved to be one of the best solutions, due to the high reprogramming efficiency and non-invasive harvesting procedure from human hair roots [3].

In the last decade, several monolayer-based differentiation protocols were developed to receive ECs from hiPSCs [4,5,6,7,8,9] or ECs and pericytes at the same time [10]. Moreover, the stable microvascular network, consisting of ECs and pericytes can be obtained from hiPSCs, using the 3D-differentiation approach [11,12]. Published protocols are based on the ability of hiPSCs to be committed to the mesodermal progenitor linage by the WNT signaling activation through GSK3 inhibition (CHIR-99021 treatment) followed by bone morphogenetic protein 4 (BMP-4) stimulation [4,6,10,11,12]. Upon stimulation with forskolin [6], fibroblast growth factor 2 (FGF-2), and vascular endothelial growth factor A (VEGF-A) mesodermal progenitors differentiate to ECs and pericytes.

Although monolayer-based protocols are widely used for EC differentiation from hiPSCs, a recent study [13] demonstrated that currently used differentiation techniques require further improvements due to the low yield and significant heterogeneity of the derived ECs. Moreover, the differentiation efficiency may depend on the origin of the hiPSC line. Although published protocols proved to be efficient for hiPSCs of fibroblast [7,8,9,10], primary mesenchymal stem cell [7], or blood outgrowth endothelial cell [10] origins, to our knowledge they were not efficiently applied for keratinocyte-derived hiPSCs.

In this study, we developed an efficient protocol of EC and pericytes differentiation from keratinocyte-derived hiPSCs that includes an additional EB stage. We compared the new EB-based protocol with one of the most commonly used monolayer-based differentiation protocols [6]. Two tested hiPSC lines efficiently differentiated into ECs and pericytes. Moreover, our EB-based differentiation protocol yielded a substantially increased number of ECs that additionally express higher levels of EC markers, such as CD31 and vascular endothelial cadherin (VE-cadherin). Our co-culture experiments demonstrated that differentiated ECs and pericytes may fulfill their main function, i.e., the blood vessel formation. Moreover, we showed that differentiated cells can be efficiently used in microfluidic devices and serve as a source for creating a 3D microvascular network.

## 2. Materials and Methods

### 2.1. hiPSC Culture

Keratinocyte-derived hiPSCs were obtained from healthy donors (k3: male, k5: male) [14] and were grown on Geltrex coating (Thermo Fisher, Waltham, MA, USA, Cat. No: A1413301) and in Essential 8™ Medium (Thermo Fisher, Cat. No: A1517001) in a humidified incubator at 37 °C with 5% CO_2_ and 5% O_2_. Subculturing was done approximately every 4 days with Versene Solution (Thermo Fisher, Cat. No: A15040066).

### 2.2. Differentiation of hiPSC-Derived ECs and Pericytes

The first 6 days of the differentiation protocol was adapted from a previously described protocol [11] with changes. On Day-1, hiPSC were singularized using TrypLE (Thermo Fisher, Cat. No: A12604021) and 4 × 10^5^ cells were mixed with 6 mL of Aggregation Medium (DMEM-F12 with GlutaMAX, 20% KnockOut Serum Replacement, 0.2% GlutaMAX, 1% Non-Essential Amino Acids, 1% Antibiotic-Antimycotic, 35 µM 2-mercaptoethanol, all from Thermo Fisher) and 50 µM of ROCK-inhibitor (Stem Cell Technologies, Vancouver, BC, Canada, Cat. No: Y-27632) and cultured in non-tissue culture treated 6-well-plates for one day. Then, the medium was changed to 6 mL of N2B27 medium (1:1 DMEM-F12 with GlutaMAX:Neurobasal Medium, 0.5% GlutaMAX, B27 Supplement without Vit. A 50×, N2 Supplement 100×, 1% Antibiotic-Antimycotic, 35 µM 2-mercaptoethanol, all from Thermo Fisher) supplemented with 12 µM CHIR99021 (Selleckchem, Munich, Germany, Cat. No: S2924) and 30 ng/mL BMP-4 (Peprotech, Rocky Hill, NJ, USA, Cat. No: 120-05) for the induction of mesoderm. On Day-3, the media was changed to 6 mL of N2B27 medium with 100 ng/mL of VEGF-A (Peprotech, Cat. No: 100-20) with 2 µM Forskolin (Sigma-Aldrich, St. Louis, MO, USA, Cat. No: F6886). On Day-5, aggregates were cut to pieces of around 200µm in diameter and plated on 0.2% gelatin-coated plates (Sigma-Aldrich, Cat. No. 9000-70-8) containing StemPro-34 medium (StemPro-34 SFM Basal Medium, StemPro-34 Nutrient Supplement 40×, 1% GlutaMAX, 1% Antibiotic-Antimycotic, 15% FBS, all from Thermo Fisher) supplemented with 100 ng/mL VEGF-A and 100 ng/mL of FGF-2 (Peprotech, Cat. No: 100-18B). On D8, medium was replaced with StemPro-34 medium supplemented with 100 ng/mL VEGF-A and 100 ng/mL of FGF-2. On Day-9 or Day-10, cells were magnetically sorted using CD31 MicroBead Kit (Miltenyi Biotec, Bergisch Gladbach, Germany, Cat. No: 130-091-935), according to the manufacturer’s instructions. CD31^+^ from the magnetic sorting gave rise to endothelial cells, whereas CD31^−^ fraction were grown with the same medium and coating. After sorting, cells were plated on 0.2% gelatin-coated plates (10 × 10^3^–13 × 10^3^ cells/cm^2^ for endothelial cells and 10 × 10^3^ cells/cm^2^ for pericytes) in Endothelial Cell and Pericyte Medium (Human Endothelial EC-SFM and 1% Antibiotic-Antimycotic from Thermo Fisher, 1% Platelet-poor Plasma Derived Serum from, from Alfa Aesar, Heysham, UK, Cat. No: J64483) with 30 ng/mL VEGF-A, 20ng/mL FGF-2 and 10 µM Y-27632 and reached confluency within 3–4 days. Medium was changed the day after sorting and every 3 days. Endothelial cells were cultured up to 5 passages, whereas pericytes could be cultured up to 8 passages using a ratio of 1:3–1:4. AccuMAX solution (Sigma, Cat. No: A7089) was used for subculturing both endothelial cells and pericytes. hiPSC-derived endothelial cells and pericytes can be cryopreserved in the culture medium with 10% DMSO and 50% FBS and should be thawed without centrifugation.

### 2.3. Immunocytochemistry

Cells were grown on 20 mm (in diameter) glass coverslips. Fixation was done with 4% paraformaldehyde for 20 min at room temperature. After washing with DPBS three times, they were blocked and permeabilized with blocking solution (10% normal donkey serum including 0.2% Triton-X-100) for 1 h at room temperature. Primary antibodies (CD31: ab9498, Abcam, Cambridge, UK; PDGFRβ: 3169, Cell Signalling Technology, Danvers, MA, USA; Claudin-5: 35-2500, Invitrogen, Waltham, MA, USA; VE-cadherin: 2158, Cell Signalling Technology; ZO-1: 33-9100, Thermo Fisher Scientific; Phalloidin: A22287, Invitrogen; CD146: 361001, Biolegend, San Diego, CA, USA; CD34: ab81289, Abcam; NG2: ab5320, Chemicon, Temecula, CA, USA; SMA: ab5694, Abcam) were diluted in blocking solution and applied overnight at 4 °C. After washing three times with DPBS, secondary antibodies (Donkey anti-Mouse IgG Secondary Antibody Alexa Fluor 488 polyclonal, Cat. No: R37114; Donkey anti-Rabbit IgG Secondary Antibody Alexa Fluor 568 polyclonal, Cat. No: A10042 from Thermo Fisher) diluted in blocking solution were applied for 2 h at room temperature. Following the washing steps, nuclei were stained with 1 µg/mL DAPI for 20 min and mounted with ProLong™ Gold Antifade Mountant (Thermo Fisher Scientific, P36931).

### 2.4. Microscopy

Phase-contrast images were taken using the EVOS FL Imaging System. For fluorescence images and image stacks, Imager.M2 Apotome1 (Carl Zeiss, Oberkochen, Germany) microscope was used.

### 2.5. Flow Cytometry

Cells were singularized using TrypLE, strained through 70 µm and 40 µm nylon meshes respectively, counted and fixed with 1% paraformaldehyde for 10 min at 37 °C. After that, they were permeabilized with 90% Methanol for 15 min on ice and stored at −20 °C.

On the day of analysis, cells were centrifuged at 300× *g* for 3 min, washed with DPBS, and incubated with primary antibodies diluted in FACS Buffer (DPBS with 0.5% BSA) for 1 h at room temperature. Cells were resuspended in FACS Buffer and subjected twice to centrifugation at 300× *g* for 3 min before being incubated with secondary antibodies (Donkey anti-Mouse IgG Secondary Antibody Alexa Fluor 488 polyclonal, Cat. No: R37114; Donkey anti-Mouse IgG Secondary Antibody Alexa Fluor 647 polyclonal, Cat. No: A-31571). Compensation particles (Anti-Mouse Ig, κ/Negative Control Compensation Particles Set, 552843, BD Biosciences, Franklin Lakes, NJ, USA) were used as negative controls. Samples were analyzed using BD LSRFortessa™ Flow Cytometer (BD Biosciences) and the data analysis was done using Flowing Software.

### 2.6. Gene Expression Analysis Using Fluidigm qRT-PCR

Total RNA isolation and gene expression analysis were performed as previously described [15]. For quantification of the gene expression of the genes of interest, Taqman assays were purchased from Thermo Fisher Scientific, USA. List of primers can be found in Appendix A.

### 2.7. Tube Formation Assay

96-well culture plates were coated with 75 µL of Matrigel (Corning, Corning, NY, USA, Cat. No: 3562319) and kept at room temperature for 10 min, and then 30 min at 37 °C incubator to allow the gelation. Endothelial cells were detached with AccuMAX solution and seeded on the Matrigel-coated plates at a density of 25 × 10^3^ cells/cm^2^ and 125 µL/well. The plates were incubated at 37 °C for 16 h and were washed once with DPBS and stained with Calcein AM (Thermo Fisher Scientific, Cat. No. C1430) for 30 min at 37 °C incubator. After the incubation period, the plates were washed once again with DPBS and, tubule formation was visualized using EVOS FL Cell Imaging System (Thermo Fisher Scientific).

### 2.8. Co-Culture Assay

EC-pericytes co-culture assay was carried out according to the previously described protocol [16], with minor modifications. 12 × 10^3^ endothelial cells and 5 × 10^4^ pericytes were plated on 0.2% gelatine-coated 96-well-plates in Endothelial Cell and Pericyte Medium with 30 ng/mL VEGF-A, 20 ng/mL FGF-2. Medium was changed in Day-1 and Day-4 along with the additional 10 µM SB431542 (Selleckchem, Cat No: S1067) supplementation. On Day-7, EC-pericyte co-culture was fixed with 4% paraformaldehyde, and immunostaining was carried out.

### 2.9. Seeding Cells in the Microfluidic System

In the project, 3D Cell Culture Chips (AIM Biotech, Nucleos, Singapore, Cat No: DAX01-1PAK) were used. 12 × 10^4^ ECs and 24 × 10^3^ pericytes were embedded in 10 µL of fibrin gel (fibrinogen: F8630, Sigma-Aldrich; thrombin, T4648, Sigma-Aldrich), following the manufacturer’s instructions. The gel was loaded in the gel chamber of the chip and kept at 37 °C for 60 min to allow the polymerization. The chip was kept in the cell culture incubator (at 37 °C, 5% CO_2_) for 7 days and the medium was replaced daily with 100 µL of Endothelial Cell Medium with 30 ng/mL VEGF-A, 20 ng/mL FGF-2.

### 2.10. Western Blot Analysis

Electrophoretic separation of the proteins was performed with the Precast Gel System (Bio-Rad, Hercules, CA, USA), 55 mg of total protein were loaded on gels. Proteins were transferred to the nitrocellulose membrane by semi-dry blotting method. The membrane was blocked in 5% (*w*/*v*) non-fat milk powder in TBS-T for 1 h at room temperature and incubated with primary antibodies ICAM-1 (Abcam, Cat No: ab53013), CD31 (Abcam, Cat No: ab9498) for 1 h at room temperature and then 1 h with secondary antibodies (IRDye, LI-COR, Lincoln, NE, USA). After each antibody incubation, the membrane was washed three times for 5 min with TBS-T. The signal was detected and evaluated by the Image Studio software (LI-COR).

### 2.11. Characterization of Microvascular Parameters

To measure vessel lateral and transverse diameter, a xyz-representation of an image stack (taken with a Zeiss Apotome) of a CD31 staining was used. Diameters were calculated using the “Plot profile” tool of ImageJ (https://imagej.net/Fiji) and measuring the distance of the CD31^+^ vessel walls CD31. Here, the maximum intensity of the CD31^+^ signal in the vessel walls were taken as reference points as indicated in the respective graphs. The tube formation assay and the 2D co-culture assay were analyzed with “Angiogenesis Analyzer” tool of ImageJ [17].

### 2.12. Statistical Analysis

Data were analyzed using two-sided Student’s t-test. A *p*-value of < 0.05 was considered statistically significant.

## 3. Results

### 3.1. Generation of ECs and Pericytes from Keratinocyte-Derived hiPSCs

Based on previously published microvascular organoid differentiation [11], we developed a multistep differentiation protocol that successively induced the mesoderm specification followed by EC and pericyte differentiation of aggregated hiPSCs (Figure 1a) and yielded a monolayer of pure EC and pericyte population after magnetic-activated cell sorting (MACS) (Appendix A).

In the first step of differentiation, hiPSCs were aggregated and EBs were formed (Figure 1b). Mesoderm differentiation was induced on day 0 via activation of WNT-signaling, by adding CHIR-99021 and BMP-4 to the medium. The addition of forskolin and VEGF-A to the medium on day 3 triggered the endothelial differentiation.

To increase the yield of ECs and pericytes, cell aggregates were transferred to gelatine-coated culture plates with subsequent growth attached at day 5 of differentiation (Figure 1b). Those conditions promoted cell propagation and further allowed the efficient transition from 3D to 2D culture conditions. The ideal size of aggregates (150–250 μm) was critical for cell propagation and differentiation, as too large colonies prevented cell spreading and suppressed differentiation, while inducing cell death (data not shown). Therefore, aggregates were reduced in size by pipetting or cutting with micro scissors.

On day 6, colonies of proliferating ECs could be observed (Figure 1b). Cells were grown until reaching confluency (at day 10), collected, and sorted. ECs could be separated from pericytes by magnetic-activated cell sorting (MACS) or fluorescence-activated cell sorting (FACS), yielding CD31^+^ (ECs) cells from CD31^−^ (pericytes).

### 3.2. Characterization of hiPSC-Derived ECs and Pericytes

ECs were sorted with MACS, based on CD31 expression. The sorted cells showed elongated morphology (Figure 1c) and expression of the EC markers CD31, VE-cadherin, CD146, and CD34 (Figure 2a, Appendix A). Moreover, those cells demonstrated the expression of claudin-5 and zonula occludens 1 (ZO-1) proteins, a clear indication of tight junction formation (Figure 2a).

To test the functionality of ECs, we performed a tube-formation assay. To this end, ECs were plated on Matrigel and stained with fluorescent dye Calcein for visualization. After 16 h, hiPCS-derived ECs formed tube-like structures on Matrigel (Figure 2b, Appendix A), suggesting that ECs were functional and capable of forming blood vessels. Additionally, we tested one of the essential properties for modelling diseases in vitro in ECs—the response to proinflammatory stimulus [18]. We exposed ECs to the tumor necrosis factor alpha (TNF-α). After 20 h, TNF-α strongly induced the expression of the intercellular adhesion molecule 1 (ICAM-1) (Appendix A).

In contrast, cells derived from the CD31^−^ fraction appeared larger than the CD31^+^ (Figure 1c) endothelial cells and expressed the pericyte markers Platelet Derived Growth Factor Receptor β (PDGFR-β), Neural/glial antigen 2 (NG2), as well as smooth muscle actin a (SMA) (Figure 2c, Appendix A).

To confirm these results, the mRNA levels of genes commonly expressed in endothelial cells or pericytes were determined using qPCR. Our findings indicate that ECs displayed a significantly higher expression of the endothelial marker genes, endothelial tyrosine kinase (*TEK*), VEGF receptor 2 (*ANGPT1*), CD31 (*PECAM1*), CD34, VE-cadherin (*CDH5*), and CD146 (*MCAM*) (Appendix A), whereas pericytes were characterized by higher expression of their respective markers, *PDGFRB* and NG2 (*CSPG4*) (Appendix A).

### 3.3. Comparison of the EB-Based with a Monolayer-Based Differentiation Protocol

To test the reproducibility of the differentiation protocol, we tested the differentiation efficiency from two different keratinocyte-derived hiPSC lines that had been reprogrammed by transduction with lentivirus. The differentiation protocol was efficient and provided reproducible results for two independent lines (k3 and k5) (Figure 3). Additionally, we compared the differentiation efficiency of the EB-based differentiation protocol with a published monolayer protocol [6]. All two differentiated hiPSC lines could be efficiently differentiated into ECs and pericytes with the use of both EB-based and monolayer-based protocols. However, the EB-based differentiation protocol yielded a higher amount of CD31^+^ and VE-cadherin^+^ ECs compared to the monolayer-based protocol (Figure 3). Interestingly, the population of CD31^+^ ECs was far more distinct displaying higher levels of CD31 and VE-cadherin. Altogether, these results demonstrate that the EB-based differentiation protocol is reproducible and suitable for different keratinocyte-derived hiPSCs lines.

### 3.4. Co-Culture of hiPSC-Derived ECs and Pericytes as a Functional Assay

To test the ability of hiPSC-derived ECs and pericytes to interact and to form blood vessel-like structures in vitro, we performed a previously described co-culture assay [16]. The assay is based on the ability of co-cultured ECs and pericytes to recapitulate the initial stages of primary vascular plexus formation.

Therefore, cells were cultured in media supplemented with VEGF-A, FGF-2, and the transforming growth factor beta (TGF-β) inhibitor SB431542. On day 7, ECs were visualized by immunofluorescence staining with CD31, while pericytes were visualized by PDGFR-β expression (Figure 4a). CD31-positive ECs surrounded by pericytes formed vessel-like structures.

As hiPSC-derived ECs successfully formed vessel-like structures on the cell culture plates, we tested their functionality in the commercially available 3D microfluidic systems (AIM Biotech), which is widely used to study the blood vessel formation and function [19,20]. ECs and pericytes were embedded in a 3D fibrin gel supplemented with VEGF-A and FGF-2 for 7 days. Starting from day 4, ECs and pericytes generated a vascular network, which continued to expand until day 7 (Figure 4b). CD31^+^ ECs were aligned and formed the branching vessel-like structures, while surrounded by CD31 negative pericytes (Figure 4c). To confirm that the microvascular network also formed lumens, we took z stack images of the chip and measured the lateral and traverse diameter at a position (Appendix A). Here, we could observe that the vessel was continuous and had a lateral diameter of 63.5 µm and a traverse diameter of 31.3 µm.

These results demonstrate that both hiPSC-derived ECs and pericytes are able to fulfill their main functions, i.e., the formation of blood vessels. Therefore, the EB-based differentiation protocol can be successfully used to model cardiovascular disorders as well as for drug testing and translational medicine approaches.

## 4. Discussion

ECs and pericytes differentiated from hiPSCs [4,6,8,9] can be employed for disease modelling and personalized medicine approaches. However, a recent large-scale single-cell RNA-Seq study revealed that the currently used monolayer based differentiation protocols can be inefficient and requires further optimization [13]. Based on our experience, monolayer-based EC differentiation protocols are also not always very efficient using keratinocyte-derived hiPSCs lines. On the other hand, recently published studies involving 3D human blood vessel organoid (VO) differentiation [11,12] demonstrated a high efficiency and yielded microvascular networks consisting of ECs and pericytes. Although VOs provide a valuable model system, some approaches, such as microfluidic chip technology as well as transwell studies, still require a tight monolayer of ECs and pericytes. Unfortunately, VOs did not prove to be an optimal source of vascular cells, due to the limited amount of cells, elevated death during cell singularization and the switch from 3D to 2D cultivation (data not shown). Given these challenges, we aimed at developing a new robust differentiation protocol by employing mesoderm induction and early endothelial specification of hiPSC aggregates under 3D conditions (EB stage), followed by propagation and maturation of ECs and pericytes attached to the culture plates (adherent stage). Thus, we took advantage of the 3D differentiation protocols which frequently provide conditions closer to the embryonic development and therefore an improved physiological microenvironment for differentiation, such as cell-to-cell interaction and gradients of growth factors.

Upon designing an EB-based protocol we were able to generate ECs and pericytes from keratinocyte-derived hiPSCs which can be efficiently sorted by FACS as well as magnetic-activated cell sorting (MACS) based on the expression of CD31 and VE-cadherin. The protocol demonstrated high reproducibility and efficiency (about 70% of ECs and 30% of pericytes) (Appendix A). Received cells could be successfully cryopreserved in the freezing medium containing 10% DMSO and 50% FBS (Appendix A).

We compared the EB-based protocol with one of the most efficient published monolayer-based differentiation protocols [6]. Our protocol demonstrated a higher differentiation efficiency compared to the monolayer-based protocol and yielded two distinguishable cell populations (ECs and pericytes) that could be easily sorted. Moreover, EC population was characterized by a significantly higher expression level of CD31 and VE-cadherin, making cell sorting easier and more efficient.

More importantly, the differentiation of ECs and pericytes from keratinocyte-derived iPSCs is associated with substantial vascularization, therefore fulfilling their genetically-imprinted function. To this end, we performed the in vitro co-culture assay [10]. Our results demonstrate that ECs and pericytes can form blood vessel-like structures in vitro suggesting that they can provide a physiological model for further mechanistic and pathomechanistic studies. Additionally, we showed that differentiated ECs respond by increased ICAM-1 expression to proinflammatory stimuli, such as TNF- α.

The rapidly developing field of microfluidic chip modelling employs vascular cells for personalized drug testing, tissue engineering, and disease studies. For example, microfluidic technologies are used to study the physical properties of blood brain vessels [20], tumor vascularization [21] or cancer metastasis [22]. To ensure that developed differentiation protocol yielded cells that can be used for these studies, we tested differentiated ECs and pericytes in one of the most commonly used commercial chip systems from AIM Biotech [23,24,25]. Using hiPSC-derived ECs and pericytes we were able for the first time to develop a 3D microvascular network inside the 3D gel. As one of the most important characteristics of blood vessels is the diameter, we visualized and measured the diameter of the microvascular network. The approximate lateral and transverse diameter of the lumen was 63.5 μm and 31.3 μm, respectively. Moreover, our results are in line with previous study: in this work, hiPSC-derived ECs and primary human pericytes were used to form the microvascular network in the AIM Biotech microfluidic chip, which had similar vessel diameter sizes (lateral diameter around 64 μm and transverse diameter 27 μm) to the vessels we have generated [20].

The described protocol yields around 70% of ECs and 30% of pericytes, which can be later used in the 1:5 (pericytes:ECs) ratio for the microvascular formation. Therefore, we suggest that ECs and pericytes derived by the here described protocol will provide a robust cell source for future microfluidic chip studies, including tumor vascularization, cardiovascular disease models, and drug testing. It will also allow to use widely accessible keratinocyte-derived hiPSC lines to obtain vasculature cells.

In summary, we developed an efficient differentiation protocol of hiPSCs to ECs and pericytes that was (1) reproducible for two independent keratinocyte-derived hiPSC lines; (2) more efficient compared to the monolayer-based protocol; (3) yielded both functional ECs and pericytes. The development of differentiation protocols will be of high interest for elucidating signaling pathways that direct vessel differentiation as well as help to develop personalized drug development approaches and cardiovascular disease modelling.

## Figures and Tables

**Figure 1 cells-10-00074-f001:**
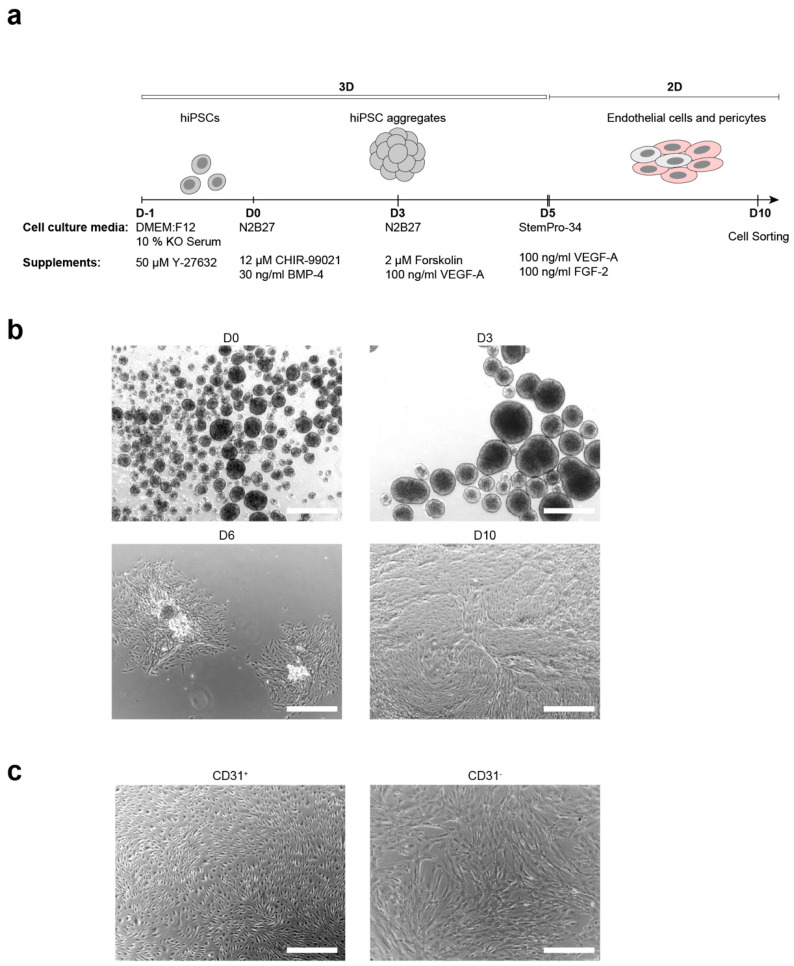
Generation of endothelial cells (ECs) and pericytes from keratinocyte-derived human induced pluripotent stem cells. (**a**). Schematic representation of the embryoid body based differentiation protocol. (**b**). Bright-field images of critical stages of differentiation: hiPSC aggregates formed at day 0 (D0), cell aggregates after mesoderm induction (D3), attached colonies at the day 6 (D6), differentiated cells before cell sorting (D10); scale bars: 500 μm. (**c**). Bright-field images that show the morphology of sorted ECs (CD31^+^) and pericytes (CD31^−^); scale bars: 500 μm.

**Figure 2 cells-10-00074-f002:**
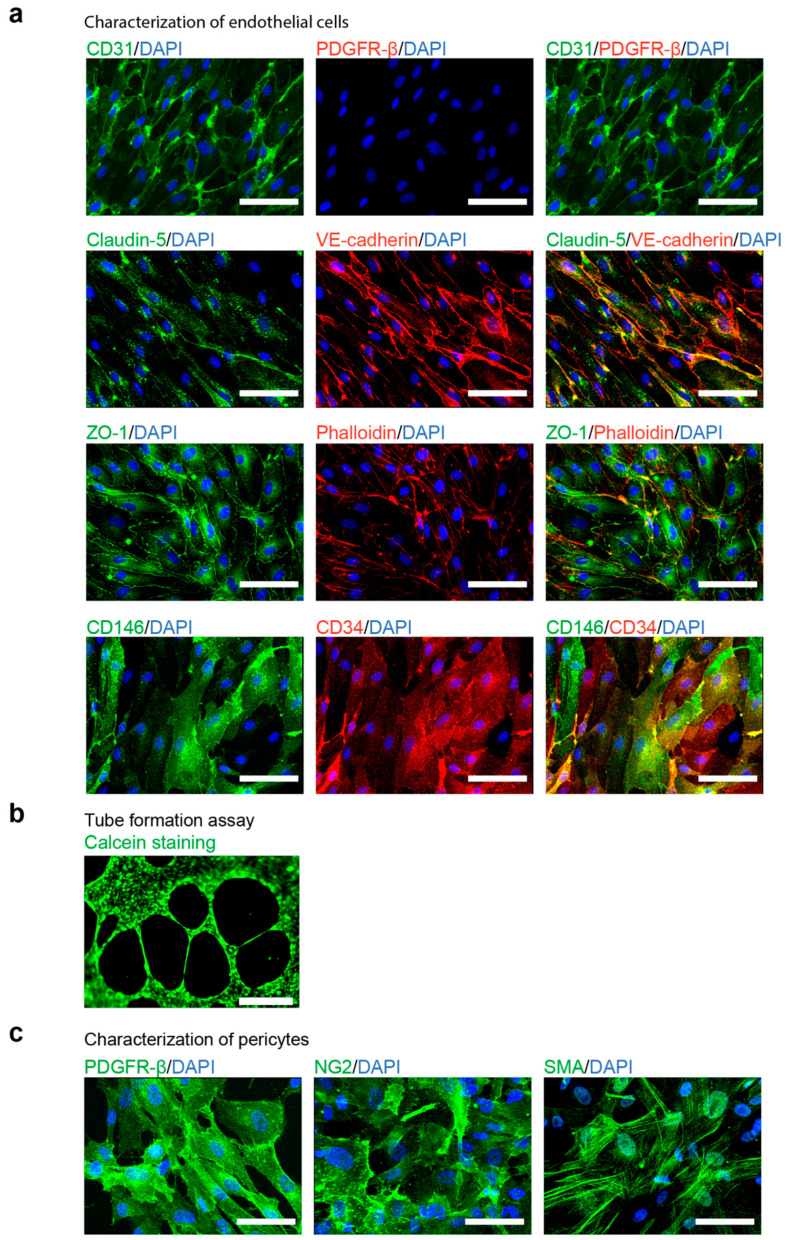
Characterization of differentiated endothelial cells (ECs) and pericytes. (**a**). Fluorescent imaging analysis of ECs positively sorted by CD31 expression with MACS system. ECs express endothelial markers CD31, vascular endothelial cadherin (VE-cadherin), CD34 and CD31 as well as components of tight junctions (zonula occludens 1 (ZO-1) and Claudin-5) but not the pericyte marker Platelet Derived Growth Factor Receptor β (PDGFR-β); scale bars: 50 μm. (**b**). In-vitro tube formation assay of ECs plated on Matrigel for 16 h; Calcein live-cell staining; scale bar, 500 μm. (**c**). Fluorescent imaging analysis of pericytes, expressing Platelet Derived Growth Factor Receptor β (PDGFR-β), Neural/glial antigen 2 (NG2) and smooth muscle actin a (SMA); scale bars: 50 μm.

**Figure 3 cells-10-00074-f003:**
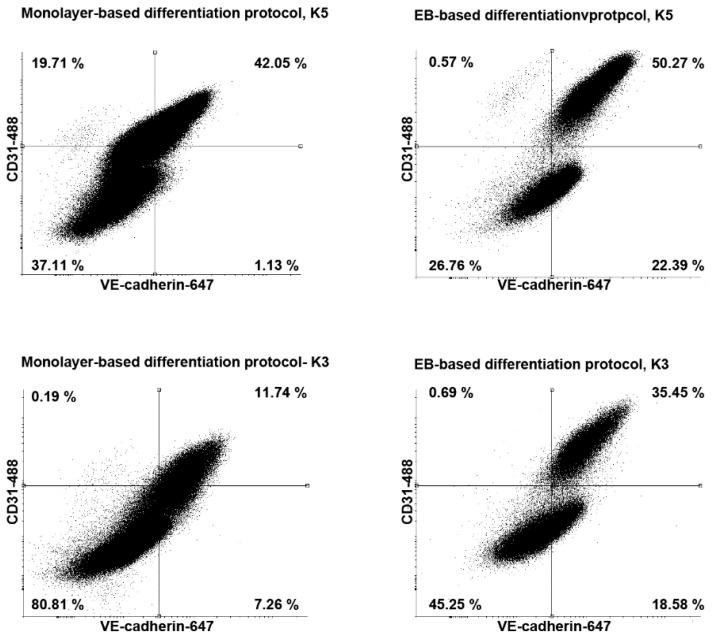
Flow cytometry analysis of CD31 and vascular endothelial cadherin (VE-cadherin) expression of endothelial cells (ECs) derived from different human induced pluripotent stem cells lines. ECs differentiated with embryoid body based protocol express higher level of VE-cadherin and CD31.

**Figure 4 cells-10-00074-f004:**
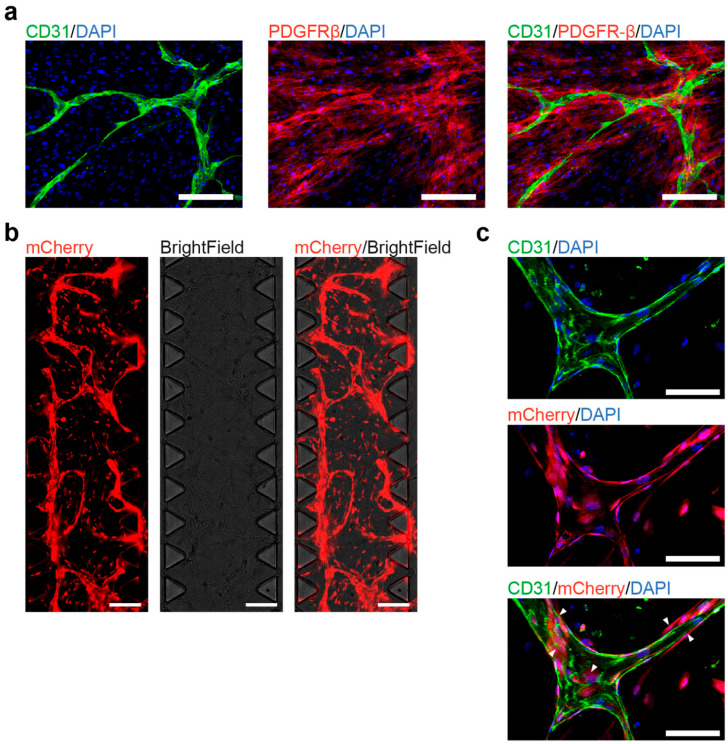
Co-culture of human induced pluripotent stem cells (hiPSC)-derived endothelial cells (ECs) and pericytes. (**a**). Co-cultured hiPSC-derived ECs and pericytes form blood vessel-like structures in vitro under 2D culturing conditions. Fluorescent imaging shows blood vessel-like structures that contain ECs (positively stained for CD31) and pericytes (positively stained for Platelet Derived Growth Factor Receptor β (PDGFR-β)); scale bars: 100 μm. (**b**). Co-cultured hiPSC-derived ECs and pericytes form a 3D microvascular network in the chip. Live-cell imaging, both ECs, and PCs express mCherry for visualization; scale bars: 1 mm (**c**). Fluorescent imaging shows the 3D microvascular network in the chip, ECs and PCs express mCherry, ECs are positively stained for CD31. White arrows show the colocalization of pericytes with ECs; scale bars: 100 μm.

## Data Availability

All data are included in the paper. There are no databases associated with this manuscript.

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
