# Peer review of "Generation of Functional Vascular Endothelial Cells and Pericytes from Keratinocyte Derived Human Induced Pluripotent Stem Cells"

_cells, 2021, doi:10.3390/cells10010074_

Round 1

Reviewer 1 Report

In this study, Pars and collaborators describe a protocol allowing the conversion of keratinocytes derived human induced pluripotent stem cells into vascular endothelial cells and pericytes.

Recent advances in the field of pluripotent stem cells have fueled new perspectives in term of regenerative medicine, disease modeling and drug screening. In this context, the possibility to generate vascular cells from human pluripotent stem cells, independently of their origin, has emerged as an important field of research. Consequently, in the last couple of years, a growing number of studies have been published describing different approaches to generate vascular cells from human pluripotent stem cells (Kusuma et al., 2013 ; Palpant et al., 2015 ; Palpant et al., 2017 ; Patsch et al., 2015 ; Prasain et al., 2014 ; ian et al., 2014, James et al., 2010 ; Yang et al., 2008 ; Levenberg et al., 2002 ; Wang et al., 2004). Within this frame of reference, the novelty of this study is difficult to assess even if the study is well-though.

In addition, different concerns can be raised and need to be addressed :

  1. The authors claim that the novelty of their findings result mainly in the fact that their protocol is based on the use of i). Keratinocytes derived human induced pluripotent stem cells and ii). 3D protocol.

Indeed, most of the studies describing so far a 3D EB-based protocol were using human embryonic stem cells. Nonetheless, the protocol described by Palpant et al. (Palpant et al., Nat. Protocol 2017) was using hiPSC lines and a EB step. How the authors do position their study regarding this ?

  1. Figure 2 is only descriptive and a quantification of the number of cells expressing the different markers indicated should help to evaluate the maturity of the system ? in addition, the subtypes of ECs generated should be characterized. Do the authors have an idea whether the generated ECs are from cardiogenic mesoderm or hemogenic mesorderm ? a better characterization using KDR, GATA1, RUNX1 could be used.
  2. Figure 3 : in the text, page 8, line 229, the authors mentioned three different keratinocyte-derived hiPSCs but in Figure 2, they only present data on two hiPSC lines ? Although it might be out of the scope of this study, it could be however interesting to have an idea of the reproducibility of the protocol. From figure 3, the efficiency of differentiation seems to be very heterogenous. In addition, the EB-based protocol does not seem to attenuate the vairability of differentiation when compared to mono-layer based differentiation.
  3. The functionality of hiPSC-derived ECs has been evaluated mainly by immunostaining. Confocal analysis using a Z-stack acquisition should first help to better visualize the lumen formation within the structures. The authors should add more functional assays as for example the endocytosis of LDL by hiPSC-derived ECs or the expression of ICAM after TNFa stimulation. Another important point to evaluate the functionality will reside in the capacity of these cells to develop vessels after subcutaneous transplantation. They should also compare with primary human endothelial cells to evaluate the level of maturity. Ideally, 
  4. Finally, do the protocols described here are compatible with a freezing step ? this point could be interesting as it would represent an important step for translational research.

Minor comments :

None of the references in the materials section is correctly inserted.

Author Response

Dear reviewer,

thank you for the time and effort that you dedicated to providing the feedback on the manuscript. Here please find the response to the comments.

In this study, Pars and collaborators describe a protocol allowing the conversion of keratinocytes derived human induced pluripotent stem cells into vascular endothelial cells and pericytes.

Recent advances in the field of pluripotent stem cells have fueled new perspectives in term of regenerative medicine, disease modeling and drug screening. In this context, the possibility to generate vascular cells from human pluripotent stem cells, independently of their origin, has emerged as an important field of research. Consequently, in the last couple of years, a growing number of studies have been published describing different approaches to generate vascular cells from human pluripotent stem cells (Kusuma et al., 2013 ; Palpant et al., 2015 ; Palpant et al., 2017 ; Patsch et al., 2015 ; Prasain et al., 2014 ; ian et al., 2014, James et al., 2010 ; Yang et al., 2008 ; Levenberg et al., 2002 ; Wang et al., 2004). Within this frame of reference, the novelty of this study is difficult to assess even if the study is well-though.

Answer

We are aware that the endothelial differentiation protocols have been published in the last decades, and we cited the most relevant publications in the introduction section. However, we believe that our work contributes to the field and offers some improvements.

Firstly, we employed hiPSC instead of hESCs, as in Yang et al., 2008 (H1 line) Levenberg et al., 2002 (H9 line), Wang et al., 2004 (H9, and H1 lines).

Secondly, fibroblast-derived hiPSCs were used in all mentioned works: Kusuma et al., 2013 (MR3, MMW2 lines), Palpant et al., 2015 and Palpant et al., 2017 (WTC11, and IMR90 lines), Prasain et al., 2014 (DF19-9-11T line), Lian et al., 2014 (19-9-11, 19-9-7, 6-9-9 lines). To our knowledge, our protocol is the first optimization of the differentiation protocol for keratinocyte-derived hiPSCs. Keratinocyte-derived hiPSCs are among the most accessible source of stem cells. Based on our experience, the differentiation protocols for keratinocyte-derived hiPSCs require additional optimization. For instance, we noticed that EB-based differentiation protocols are frequently more efficient.

Thirdly, with our EB-based protocol, both components of microvascular vessels (ECs and pericytes) can be differentiated at the same time.

Therefore, in the current study we provide a protocol that (1) can be used for keratinocyte-derived iPSCs; (2) give rise to the ECs and pericytes from the same origin; (3) yield cells that can be successfully used for 2D culturing as well as culturing in the 3D gel. We hope that the differentiation protocol, that provides both main components of the blood vessel formation can make the in-vitro blood vessel studies more accessible for a wide range of research groups.

In addition, different concerns can be raised and need to be addressed :

  1. The authors claim that the novelty of their findings result mainly in the fact that their protocol is based on the use of i). Keratinocytes derived human induced pluripotent stem cells and ii). 3D protocol.

Indeed, most of the studies describing so far a 3D EB-based protocol were using human embryonic stem cells. Nonetheless, the protocol described by Palpant et al. (Palpant et al., Nat. Protocol 2017) was using hiPSC lines and a EB step. How the authors do position their study regarding this ?

Answer

We consider that the reviewer mentioned the following publication: “Generating high-purity cardiac and endothelial derivatives from patterned mesoderm using human pluripotent stem cells.”, Palpant et al. Nat Protocol 12, 15–31 (2017).

In this work, the authors have used two human induced pluripotent stem cell lines (WTC11 and IMR90) derived from fibroblasts and one human embryonic stem cell line (RUES2). The authors mentioned that they have used a monolayer-based differentiation protocol to generate endothelial cells: “In this report, we use developmental cues in a monolayer differentiation approach to generate different cardiovascular subtypes, reproducibly achieving >90% purity of all lineages without sorting” (Palpant et al., 2017). In the protocol description, authors do not mention embryoid body (EB) step and describe the monolayer culturing conditions in a 24-well-plate format.

Therefore, according to our understanding, there was no EB step during differentiation.

  1. Figure 2 is only descriptive and a quantification of the number of cells expressing the different markers indicated should help to evaluate the maturity of the system ? in addition, the subtypes of ECs generated should be characterized. Do the authors have an idea whether the generated ECs are from cardiogenic mesoderm or hemogenic mesorderm ? a better characterization using KDR, GATA1, RUNX1 could be used.

Answer

In addition to immunostainings and FACS analyses, we have performed a quantitative PCR analysis (Fig. S2) that shows the expression of EC markers (such as PECAM1, TEK, ANGPT1, CD34, CDH5, ICAM, MCAM) in the sorted EC population as well as pericyte markers (PDGFRB, CSPG4, and ANPEP) in the pericyte population. We also confirmed that ECs express tight junction components (Claudin-5 and ZO1) both on the protein and mRNA level (Fig. 2a, Fig. S3). Moreover, almost all (99 %, Fig. S1) ECs expressed both CD31 and VE-cadherin markers, suggesting that ECs are mature.

However, we assume that the functional analysis of ECs and pericytes may confirm the maturity of cells. We showed that ECs are able to form tube-like structures during the tube-formation assay (Fig. 2b). Pericyte fraction was able to support the blood-vessel like structures in the primary vascular plexus formation assay (Fig. 4a). Finally, co-cultured on the 3D gel, ECs and pericytes were lined up, and formed the blood vessel-like structure, with ECs surrounded by pericytes. The microvascular network also formed the lumen.

In summary, we think that the expression of cell markers, functional assays, and the ability of cells to form microvasculature could indicate that cells are functional and mature.

As the reviewer suggested, for the revised manuscript we also assessed the ECs respond to the TNFalpha treatment. This showed an increased level of ICAM-1 expression (Fig. S2).

In the current study, we did not aim to describe the mesoderm differentiation. However, there is evidence of the combinatorial effect of Wnt agonist CHIR99021 and BMP-4 to upregulate Brachyury expression in early EC differentiation (Patsch et al., 2015). This could mean that our ECs are derived from hemogenic mesoderm. We agree that further studies of the mesoderm development can be interesting, however, as we are rather focused on the production of ECs, we have not addressed this question in our work.

  1. Figure 3 : in the text, page 8, line 229, the authors mentioned three different keratinocyte-derived hiPSCs but in Figure 2, they only present data on two hiPSC lines ? Although it might be out of the scope of this study, it could be however interesting to have an idea of the reproducibility of the protocol. From figure 3, the efficiency of differentiation seems to be very heterogenous. In addition, the EB-based protocol does not seem to attenuate the vairability of differentiation when compared to mono-layer based differentiation.

Answer

In the text, page 8, line 229 there was a typo and we corrected it; we tested two keratinocyte-derived hiPSC lines. The protocol showed a high level of reproducibility. We now included the data on the efficiency of differentiations in Table S5 and discussion (page 12, lines 314-315).

  1. The functionality of hiPSC-derived ECs has been evaluated mainly by immunostaining. Confocal analysis using a Z-stack acquisition should first help to better visualize the lumen formation within the structures. The authors should add more functional assays as for example the endocytosis of LDL by hiPSC-derived ECs or the expression of ICAM after TNFa stimulation. Another important point to evaluate the functionality will reside in the capacity of these cells to develop vessels after subcutaneous transplantation. They should also compare with primary human endothelial cells to evaluate the level of maturity. Ideally, 

Answer

As suggested, we added an additional functionality assessment of endothelial cells. We showed the increased ICAM -1 production after TNFalpha stimulation (Fig. S2). We added the corresponding data to section 2.10 Western Blot analysis, 3.2 Characterization of hiPSC-derived ECs and pericytes (page 7, lines 223-226), and added Fig. S2.

As suggested, the we visualized and measured the lumen of the microvascular: a Z-stack acquisition was added, which demonstrates that the 3D microvasculature resembles the blood vessel structures. We added the lumen visualization to Figure S3. We also changed section 2.11 Characterization of microvascular parameters, 3.4 Co-culture of hiPSC-derived ECs and pericytes as a functional assay (page 10, lines 275-278), discussion (page 12, lines 337 – 343).

Although we agree that the functional comparison to primary cells will provide valuable information, we aimed in the study to provide a robust differentiation protocol that could serve as an alternative to already published protocols and will allow receiving ECs and pericytes in the same step and did not address the maturation stages of ECs.

As subcutaneous transplantation requires animal experiments and therefore arise ethical question we tried to avoid them and rather replace with in vitro assessments, such as primary vascular plexus formation assay that was previously used to demonstrate the functionality of hiPSC-derived ECs and pericytes (Orlova et al., 2013) and microvascular formation on the microfluidic device.

  1. Finally, do the protocols described here are compatible with a freezing step ? this point could be interesting as it would represent an important step for translational research.

Answer

The description of the freezing step can be found in the Methods Section 2.2. To emphasize the possibility to freeze both endothelial cells and pericytes we now included the table in the supplementary material Table S4 that demonstrates cell viability after thawing and mentioned it in the discussion (page 12, lines 315-317).

Minor comments :

None of the references in the materials section is correctly inserted.

Answer

We are sorry for the inconvenience, the corrections were made.

Reviewer 2 Report

This manuscript is a short description of a modified protocol to produce endothelial cells (ECs) and pericytes from human keratinocyte-derived iPSCs. The authors show yields in comparison to previously published protols and some functional assays such as EC tube formation and pericyte marker expression. They further show that these differentiated ECs and pericytes can be co-cultured on commonly used microfluidic devices.  The manuscript is technically sound; however, there are some issues regarding the rationale, comparison to previous protocols and discussion of the results as outlined below.

  1. The images are beautiful; however, the manuscript generally lacks quantitative analysis of images, including for the tube formation assay, the characterization of pericytes and the microfluidic system (Fig. 2, 4).

  1. The major claim of this manuscript is that a two-stage differentiation protocol of hiPSCs is more efficient and faster. However, comparison to ‘traditional’ monolayer-based differentiation is only shown in higher yields of CD31+ and VE-cadherin+ ECs and not for functional outcomes. Do higher yields in CD31 and VE-cadherin+ ECs induce any changes in functionality and marker expression shown in Fig. 2? Beyond yield, can the authors quantify and discuss some additional parameters such as time, cost, etc? This seem to be required to make this protocol useful for other researchers and applications and to further support the advantage of EB-based differentiation protocols.

  1. The use of these cells for co-culture in a microfluidic device is interesting but the relevance, especially with regard to the differentiation protocol is less clear. The authors discuss potential applications of this culture device, but no functional outcome or testing is shown in the manuscript. Here, again, some quantitative measures and comparisons to previous protocols and/or studies would be useful.

Author Response

Dear reviewer,

thank you for the time and effort that you dedicated to providing the feedback on the manuscript. Here please find the response to the comments.

This manuscript is a short description of a modified protocol to produce endothelial cells (ECs) and pericytes from human keratinocyte-derived iPSCs. The authors show yields in comparison to previously published protols and some functional assays such as EC tube formation and pericyte marker expression. They further show that these differentiated ECs and pericytes can be co-cultured on commonly used microfluidic devices.  The manuscript is technically sound; however, there are some issues regarding the rationale, comparison to previous protocols and discussion of the results as outlined below.

  1. The images are beautiful; however, the manuscript generally lacks quantitative analysis of images, including for the tube formation assay, the characterization of pericytes and the microfluidic system (Fig. 2, 4).

Answer

We now included the quantification analysis of images for Fig. 2 and Fig. 4 in the Supplementary Materials, Table S1, S2, and S3.

  1. The major claim of this manuscript is that a two-stage differentiation protocol of hiPSCs is more efficient and faster. However, comparison to ‘traditional’ monolayer-based differentiation is only shown in higher yields of CD31+ and VE-cadherin+ ECs and not for functional outcomes. Do higher yields in CD31 and VE-cadherin+ ECs induce any changes in functionality and marker expression shown in Fig. 2? Beyond yield, can the authors quantify and discuss some additional parameters such as time, cost, etc? This seem to be required to make this protocol useful for other researchers and applications and to further support the advantage of EB-based differentiation protocols.

Answer

The higher level of CD31 and VE-cadherin will be beneficial because it facilitates cell sorting procedures. In particular, the magnetic beads sorting (MACS) is not efficient for the 2D protocol. The data with the efficiency of MACS is now included in the supplementary material (Table S5). The MACS can be beneficial because it does not require FACS and therefore can be more accessible for many laboratories. FACS sorting of a distinct population of cells is also easier and can help to increase the purity of the sorted population.

The main advantage of the protocol is the ability to receive both endothelial cells and pericytes at the same time during differentiation (about 71% of endothelial cells and 29% of pericytes). Both cell populations are used for the microvasculature formation, in a 1:5 ratio. That makes the differentiation protocol very efficient for those, who aim to use the microvasculature in the 3D gels for their studies, therefore we hope that our protocol will make the hiPSC-derived endothelial cells and pericytes more accessible.

We included Table S5 in the supplementary material to show the higher efficiency of the EB-based protocol. We also changed the discussion section (page 12, lines 314-317 and 344-345) to make it clear. 

  1. The use of these cells for co-culture in a microfluidic device is interesting but the relevance, especially with regard to the differentiation protocol is less clear. The authors discuss potential applications of this culture device, but no functional outcome or testing is shown in the manuscript. Here, again, some quantitative measures and comparisons to previous protocols and/or studies would be useful.

Answer

We tested the ability of endothelial cells and pericytes to form a 3D microvascular network in the microfluidic device to confirm the functionality of both cell types. To our knowledge, we described for the very first time that hiPSC-derived pericytes can be used in the microfluidic for 3D microvascular formation. For this reason, we could not compare our results with previously published microvascular systems.

However, we now added the analysis of lumen formation and quantified the lumen size – one of the main characteristics of blood vessels. This data will allow comparing our system with a microvasculature network formed by primary cells and/or hiPSC derived cells. We added Figure S3 and changed the discussion section (page 10, lines 275-278).

Moreover, several working groups reported that they have difficulties with using hiPSC-derived endothelial cells for microfluidic devices. We aimed to demonstrate that our protocol received cells that can be successfully cultured in one of the most commonly used microfluidic systems.

In further studies we and other groups will use the microfluidic systems for drug testing and disease modeling, therefore the ability to receive blood vessels from the same hiPSC source can be extremely valuable for translational research.

Round 2

Reviewer 2 Report

The authors adequately addressed my questions with additional quantifications and clarifications in the discussion. I have no further concerns.